# INTERACTIVE PORTRAIT HARMONIZATION

**Jeya Maria Jose Valanarasu**[1]\*, **He Zhang**[2], **Jianming Zhang**[2], **Yilin Wang**[2], **Zhe Lin**[2],
**Jose Echevarria**[2], **Yinglan Ma**[2], **Zijun Wei**[2], **Kalyan Sunkavalli**[2], **Vishal M. Patel**[1]
1 Johns Hopkins University, 2 Adobe Research

## ABSTRACT

Current image harmonization methods consider the entire background as the guidance for harmonization. However, this may limit the capability for user to choose any specific object/person in the background to guide the harmonization. To enable flexible interaction between user and harmonization, we introduce interactive harmonization, a new setting where the harmonization is performed with respect to a selected *region* in the reference image instead of the entire background. A new flexible framework that allows users to pick certain regions of the background image and use it to guide the harmonization is proposed. Inspired by professional portrait harmonization users, we also introduce a new luminance matching loss to optimally match the color/luminance conditions between the composite foreground and select reference region. This framework provides more control to the image harmonization pipeline achieving visually pleasing portrait edits. Furthermore, we also introduce a new dataset carefully curated for validating portrait harmonization. Extensive experiments on both synthetic and real-world datasets show that the proposed approach is efficient and robust compared to previous harmonization baselines, especially for portraits. The code can be found here: https://github.com/jeya-maria-jose/Interactive-Portrait-Harmonization

## 1 INTRODUCTION

With the increasing demand of virtual social gathering and conferencing in our lives, image harmonization techniques become essential components to make the virtual experience more engaging and pleasing. For example, if you cannot join a wedding or birthday party physically but still want to be in the photo, the first option would be to edit yourself into the image. Directly compositing yourself into the photo would not look realistic without matching the color/luminance conditions. One possible solution to make the composition image more realistic is to leverage existing image harmonization methods Cong et al. (2020; 2021); Cun & Pun (2020); Ling et al. (2021); Jiang et al. (2021); Guo et al. (2021b); Tsai et al. (2017).

Most previous works focus on a more general image harmonization setup, where the goal is to match a foreground object to a new background scene without too much focus on highly retouched portrait. However, when we conduct surveys among professional composition Photoshop/Affinity [1] users, we realized that portrait harmonization is the most common task of image editing in real-world scenario and professional settings. This makes portrait harmonization the most important use case of image harmonization. We note that previous harmonization works have not focused on addressing portrait harmonization on real-world data. In this work, we aim to explore a better solution to obtain realistic and visually pleasing portrait harmonization for real-world high-resolution edited images.

One common question that pops up when we demo existing image harmonization workflow to these professional users is: '*How could we choose a certain person as reference when we do harmonization with existing workflow ?*'. The workflow design of existing state-of-the-art harmonization methods Cong et al. (2020; 2021); Ling et al. (2021); Guo et al. (2021b) limits the capability for user to choose any person/region as reference during the harmonization process. These frameworks are designed such that they just take in the composite image and foreground mask as input thus offering no specific way to help the user to guide the harmonization. Certain frameworks such as

---

\*Work done while doing an internship at Adobe
[1] https://affinity.serif.com/en-gb/photo/

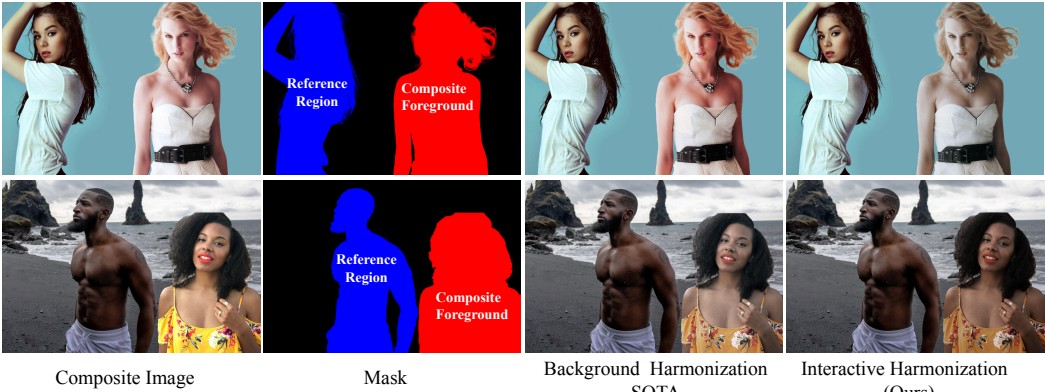

| Composite Image | Mask | Background Harmonization SOTA | Interactive Harmonization (Ours) |

Figure 1: Testing with in-the-wild portrait composites. Top row- Professional Studio Portrait, Bottom row- Non-studio portrait. It can be observed that the current SOTA harmonization method Ling et al. (2021) fails to give realistic results as it tries to match the appearance of foreground to the entire background. Our proposed interactive harmonization framework produces a visually pleasing portrait harmonization as it matches the appearance of the original portrait in the reference region instead of the entire background as selected by the user.

Bargain-Net Cong et al. (2021) have the flexibility to be tweaked and converted to serve interactive harmonization. However, from our experiments we find that they are not robust enough to perform realistic portrait harmonization.

Furthermore, professional portraits are mostly shot at studios where screens constitute the background. These screens offer little to no information in harmonizing a new composite when edited into the photo. This causes current harmonization methods to produce unstable results (see first row of Figure 1) as they have not been trained to perform harmonization with screens as background. Also, portraits which are captured by everyday users usually contain a background of spatially varying luminance characteristics. Using the entire background to guide harmonization here can result in undesirable outcomes (see second row of Figure 1) as harmonization depends on the location where the foreground is composited to.

To this end, we introduce *Interactive harmonization*- a more realistic and flexible setup where the user can guide the harmonization. The user can choose specific regions in the background as a reference to harmonize the composite foreground. From a professional stand-point, this is a much-needed feature as it could enable a lot of control for editors. In this work, we propose a new interactive portrait harmonization framework that has the flexibility to take in a reference region provided by the user. We use an encoder-decoder based harmonization network which takes in the foreground composite region as input and a style encoder which takes in the reference region as its input. The reference region to guide the harmonization is selected by the user and can be a person, and object or even just some part of the background. However, it can be noted that in portrait editing it is very common to choose another person in the picture as reference to obtain an effective harmonization. We also use a style encoder that extracts the style code of the reference region and injects it to the harmonized foreground. We carefully align the style information with foreground while also preserving the spatial content using adaptive instance normalization Huang & Belongie (2017) layers at decoder. To make the harmonization look more realistic it is important to optimally match the luminance, color, and other appearance information between the reference and foreground. To match these characteristics in manual photo editing, professional photography users usually match statistics of the highest (highlight), lowest (shadow) and average (mid-tone) of luminance points between the composite foreground image and the reference region. Hence, we propose a new luminance matching loss that is inspired by professional users [2]. In the proposed loss, we match the highlight, mid-tone and shadow points between the reference region and foreground composite region.

Current publicly available datasets are developed only for background harmonization problem. So, we curate a new **IntHarmony** dataset to enable training for interactive harmonization utilizing aug-

---

[2]https://www.youtube.com/watch?v=SoWefQNcIyY&t=268s

mentations that would work well for general as well as portrait harmonization. We also introduce a new real-world testing data **PortraitTest** with ground-truth annotations collected from professional experts to benchmark the methods. Codes, datasets and pretrained models will be made public after the review process.

In summary, the following are the key contributions of this work:

- We are the first to introduce *Interactive Harmonization* and propose a new framework which has the flexibility to solve it.
- We propose a new luminance matching loss to better match the foreground appearance to the reference region.
- We curate a new synthetic training dataset as well as a professionally annotated benchmark test for portrait harmonization and demonstrate state-of-the-art results on it.
- We show that performing interactive harmonization using our proposed method results in visually pleasing portrait edits.

## 2 RELATED WORKS

Classical methods for image harmonization perform color and tone matching by matching global statistics Reinhard et al. (2001); Ling et al. (2021). Methods like Pérez et al. (2003); Tao et al. (2010) try to apply gradient domain methods to transfer the color and tone . Zhu et al. (2015) proposed the first convolutional network-based method to improve realism of composite images. The first end-to-end deep learning-based image harmonization method was proposed by Tsai et al. (2017) where an encoder-decoder based convolutional network was used. Following that Cun & Pun (2020) proposed an attention-based module to improve harmonization. Dove-Net, proposed by Cong et al. (2020) used a Generative Adversarial Network (GAN)-based method with an additional domain verification discriminator to verify if the foreground and background of a given image belongs to the same image. Cong et al. (2020) also proposed a public dataset, iHarmony4 for image harmonization. Following that, Cong et al. (2021) proposed Bargain-Net which uses a domain extractor to extract information about the background to make the harmonized foreground more consistent with the background. Sofiiuk et al. (2021) proposed a new architecture involving pre-trained classification models to improve harmonization. Attention-based feature modulation for harmonization was proposed in Hao et al. (2020). Ling et al. (2021) proposed RainNet which introduces a region-aware adaptive instance normalization (RAIN) module to make the style between the background and the composite foreground consistent. Guo et al. (2021b) proposed intrinsic image harmonization where reflectance and illumination are disentangled and harmonized separately. It can be noted that photo-realistic style transfer methods Li et al. (2018); Luan et al. (2017); Yang et al. (2019) when adopted for harmonization do not perform well as they require similar layout as explained in Jiang et al. (2021). Jiang et al. (2021) proposed a self-supervised framework to solve image harmonization. Recently, Guo et al. (2021a) proposed an image harmonization framework using transformers. Portrait relighting methods Zhou et al. (2019); Pandey et al. (2021) have also been explored to produce realistic composite images for a desired scene.

## 3 INTERACTIVE HARMONIZATION

**Setting:** Given an input composite image $C$, the goal is to generate a harmonized image $H$ that looks realistic. Composite image corresponds to the edited image where a new person is introduced into the scene. This newly introduced region is termed as composite foreground region $F$ and the scene it was introduced into is termed as background $B$. In general harmonization, we use the background $B$ to harmonize $F$. In *interactive harmonization*, we propose using a specific region $R$ of the background $B$ to guide harmonizing $F$. Note that the region $R$ is a subset of $B$ *i.e* ($R \in B$). $R$ can be any region pertaining to the background that the user wants $F$ to be consistent with.

For portrait images, an easy way to perform realistic harmonization would be to select the person/object in the reference portrait as $R$ to guide the edited person/object $F$. This will make sure that that the luminance conditions of the reference portrait image is consistent with that of the newly edited-in portrait. However, we do not hardly constrain the region to be only a portrait in the background, it can also be objects or even a part of the scene. Please note that portraits here do not only

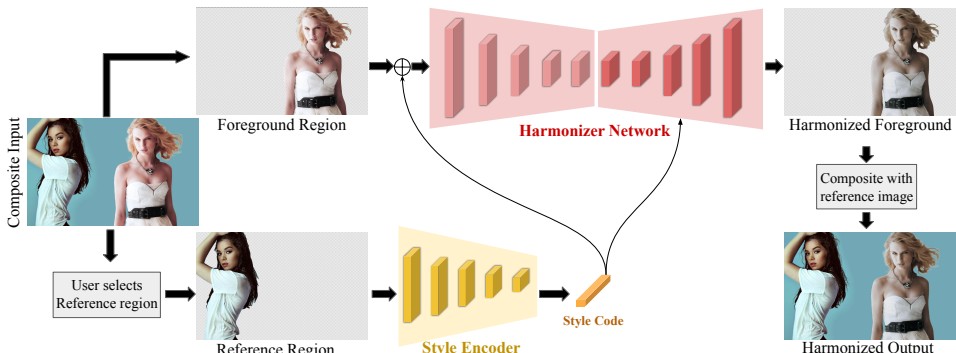

Figure 2: Overview of the proposed interactive portrait harmonization framework.

correspond to images containing people, it can also contain objects. A photo becomes a portrait when certain subjects are the main focus of the picture.

**Dataset Curation:** Publicly available datasets like iHarmony4 Cong et al. (2020) were proposed for *background harmonization* and do not provide any reference region information. So, we curate a synthetic dataset and also introduce a real-world portrait harmonization dataset for validating.

**1) IntHarmony:** This newly curated synthetic dataset specifies an additional reference region to guide the harmonization. IntHarmony has the following information for each data instance: composite image, ground truth, foreground mask of the composite foreground and a guide mask which provides information about the reference region to guide the harmonization. IntHarmony is built on top of MS-COCO dataset Lin et al. (2014). We make use of the instance masks provided in MS-COCO to simulate foreground and reference regions. First, we select a random instance mask to pick the foreground region. The selected foreground region is then augmented using a wide set of meaningful augmentations focusing on luminance, contrast and color. We use all the augmentations proposed in Jiang et al. (2021) to create the composite. More details on this can be found in the supplementary document. This augmented foreground is composited with the original image to get the composite image ($I$). Another random instance mask is used to get the reference guide mask. The original image is considered as the ground truth. The number of training images in IntHarmony is 118287 and 959 images are allocated for testing. Note that the instance masks and the augmentations are chosen at random to induce more generalizabilty to the network. IntHarmony consists of both objects, backgrounds and people and is not very focused for portraits. We use this dataset to help the network learn the interactive way to perform harmonization as we have a reference region to access. We also use a similar dataset creation scheme on Multi-Human People Parsing (MHP) dataset Zhao et al. (2018); Li et al. (2017); Nie et al. (2017) to synthesize another data further finetune the models for portrait harmonization. Note that MHP dataset contains only images with people.

**2) PortraitTest:** To validate our performance for real-time applications, we introduce a real-world test data consisting of high resolution composite portrait images. The ground truth is annotated with the help of multiple professionals using PhotoShop curve adjustment layers to change brightness, hue, saturation layer and contrast of the composite foreground to make it look more natural and pleasing. The number of images in this test dataset is 50 and the composite foreground is of varied luminance conditions, contrast and color.

## 4 METHOD

In this section, we first explain the details of the framework we propose to solve interactive harmonization. We then give the training details and the loss functions we introduce for this task.

**Network Details:** An overview of the proposed framework is shown in Figure 2. It consists of two sub-networks: a harmonizer network and a style encoder network.

**Harmonization Network** is an encoder-decoder architecture which takes in two inputs: composite foreground image ($I$), a foreground mask and a style code ($\phi$) extracted from the style encoder which is of dimension $1 \times D$. The encoder is built using 4 convolutional blocks where each conv block has a conv layer that increases the number of channels and performs downsampling, followed by a set of

residual blocks. The latent features from the encoder have a spatial dimension of $(H/16) \times (W/16)$. The decoder is also built using 4 conv blocks where each block has an upsampling layer and a set of residual blocks similar to the encoder. In addition, we use adaptive instance norm (AdaIN) layers in each conv block that takes in the style code ($\phi$) and the features from previous conv block in decoder as their input. Thus, the decoder uses the style code extracted from the reference object to guide the harmonization. The output from decoder is the harmonized foreground image. It is then composited with the original reference image to get the complete harmonized output.

**Style Encoder** takes in the reference region chosen by the user as the input. The style encoder consists of a series of partial convolutional layers Liu et al. (2018) to downsample the input image, extract meaningful features and output a style code. The partial convolution layers take in both the reference image and guide mask as their output. The latent embedding is fed through a average pooling layer to obtain the 1D style code ($\phi$).

**Training Details:** Training the network for interactive harmonization is performed stage-wise. In the first stage, we train the network to perform background harmonization, where the training data of iHarmony4 Cong et al. (2020) is used. The input to the harmonizer network is the masked out foreground image concatenated with the style code of the background extracted from the style encoder. In the next stage, we finetune our network for interactive harmonization on IntHarmony. Here, the input to the style encoder is the reference region. Thus, the network here is trained in such a way that a reference region guides the harmonization which is also due to the loss functions we introduce (see Section 4.3). Finally, we further fine-tune the network on the augmented MHP dataset for portrait harmonization to be used in professional settings. The input to the harmonizer network is the masked out foreground image (rather than entire composite image) concatenated with the style code of the background extracted from the style encoder. From our experiments, we realized that using the entire composite as input to the harmonizer network does not result in optimal training. This happens because the style code information bleeds out to the background of the composite, reducing its influence to the foreground. Masking out the background makes sure that the style code affects only the composite foreground and not the entire image.

**Loss Functions:** We use a combination of the proposed luminance matching loss, consistency loss, harmonization loss, and triplet losses for training our framework.

**Luminance Matching loss:** The main objective of interactive portrait harmonization is to make sure that the harmonized foreground matches the appearance of the reference region. To achieve this, we introduce three new losses - highlight, mid-tone and shadow losses, which are used to match the highlight, mid-tone and shadow between the reference region and foreground region to be harmonized. We define these losses as follows:

$$\mathcal{L}_{highlight} = \|\mathbf{H}_{max} - \mathbf{R}_{max}\|_1 \tag{1}$$
$$\mathcal{L}_{mid-tone} = \|\mathbf{H}_{mean} - \mathbf{R}_{mean}\|_1 \tag{2}$$
$$\mathcal{L}_{shadow} = \|\mathbf{H}_{min} - \mathbf{R}_{min}\|_1, \tag{3}$$

where $\mathbf{H}$ corresponds to the harmonized image and $\mathbf{R}$ corresponds to the reference region. These losses try to match the intensities of the foreground with the reference region at the highest, lowest and average luminescence points thus matching the contrast. For the highest and lowest points, we choose the $90^{th}$ and $10^{th}$ percentile respectively to improve stability. We define luminance matching loss ($\mathcal{L}_{LM}$) as the sum of these 3 losses:

$$\mathcal{L}_{LM} = \mathcal{L}_{highlight} + \mathcal{L}_{mid-tone} + \mathcal{L}_{shadow}. \tag{4}$$

Luminance matching loss ($\mathcal{L}_{LM}$) is illustrated in Figure 3. Note that $\mathcal{L}_{LM}$ is an main addition to our training methodology which helps to make sure that the statistics between the reference and foreground are matched. This loss penalizes the network if the statistics of the harmonized foreground and the reference region are not matched.

**Consistency loss:** We also introduce a consistency loss $\mathcal{L}_{consis}$ between style codes of the harmonized foreground and the reference region to penalize the network whenever the style code of the reference region and harmonized foreground region are different from each other. The consistency loss is defined as:

$$\mathcal{L}_{consis} = \|\phi(h) - \phi(b)\|_1. \tag{5}$$

**Harmonization loss:** We also use a generic harmonization loss which is an L1-loss between the prediction and the ground-truth as follows:

$$\mathcal{L}_{harmonization} = \|\mathbf{H} - \hat{\mathbf{I}}\|_1. \tag{6}$$

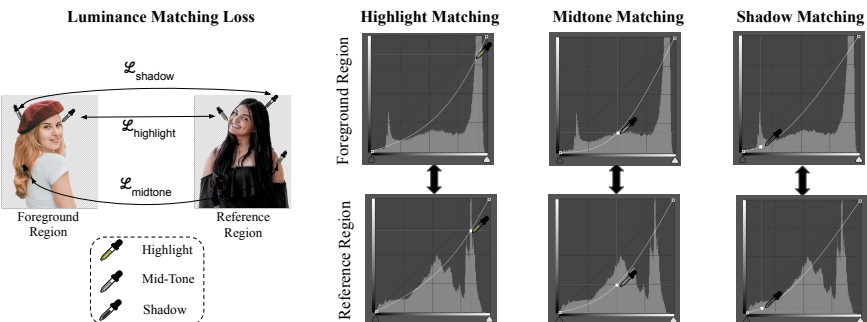

Figure 3: Overview of the proposed Luminance Matching loss.

where $\mathbf{H}$ corresponds to the harmonized image and $\hat{\mathbf{I}}$ represents the ground truth. This loss tries to minimize the distance between the prediction and input thus making sure the prediction is always consistent in appearance with that of the ground truth.

**Triplet losses:** We introduce two triplet losses similar to Cong et al. (2021) based on the style codes of the composite foreground ($c$), harmonized foreground ($h$), real foreground ($r$) (ground truth) and that of the reference region ($b$). The first triplet loss is as follows:
$$\mathcal{L}_{triplet1} = \max(\|\phi(h), \phi(b)\|_2 - \|\phi(h), \phi(c)\|_2 + m, 0).$$
This loss tries to make the style code generated by the style encoder for harmonized foreground ($h$) to be closer to that of style code of reference region ($b$) while being far away from the style code of composite foreground ($c$). $m$ here corresponds to the margin.

We define another triplet loss which forces the real foreground style code ($r$) to be close to harmonized foreground style code ($h$) while making sure the style code of harmonized foreground style code ($h$) is far away from composite foreground ($c$):
$$\mathcal{L}_{triplet2} = \max(\|\phi(h), \phi(r)\|_2 - \|\phi(h), \phi(c)\|_2 + m, 0),$$

The total loss used to train the network is as follows:
$$\mathcal{L}_{IPH} = \mathcal{L}_{harmonization} + \alpha\mathcal{L}_{LM} + \lambda\mathcal{L}_{consis} + \beta(\mathcal{L}_{triplet1} + \mathcal{L}_{triplet2})$$
where $\alpha$, $\beta$ and $\lambda$ are the factors that control the contribution the triplet losses have in the total loss.

## 5 EXPERIMENTS

**Implementation Details:** Our framework is developed in Pytorch Paszke et al. (2019) and the training is done using NVIDIA RTX 8000 GPUs. The training is done stage-wise as explained in the previous section. We use an Adam optimizer Kingma & Ba (2014) with a learning rate of $10^{-4}$, $10^{-5}$, $10^{-6}$ at each stage respectively. The batch size is set equal to 48. The images are resized to $256 \times 256$ while training. During inference, to get the high resolution images back, we use a polynomial fitting like in Afifi et al. (2019). We composite the high resolution harmonized foreground with the reference image to get the high resolution harmonized image.

**Comparison with State-of-the-art:** We compare our proposed framework with recent harmonization networks like Dove-Net Cong et al. (2020), Bargain-Net Cong et al. (2021), and Rain-Net Ling et al. (2021). We note that Bargain-Net has a domain extractor branch that directly takes in the background of the image to make the harmonization consistent with the background. In addition to comparing with the original model, we also re-train Bargain-Net in a interactive setup similar to our proposed approach for fair comparison. Here, we feed in the reference region as the input to the domain extractor and use similar protocols as our method. We call this configuration Bargain-Net (R). Note that Bargain-Net (R) is trained with the same data and stage-wise training as our method for fair comparison. We explain why the other frameworks cannot be converted for interactive harmonization in the supplementary material. We use publicly available weights of Dove-Net, Bargain-Net and Rain-Net to get their predictions on PortraitTest and IntHarmony. We denote our approach as IPH (Interactive Portrait Harmonization). While testing with IntHarmony, Bargain-Net (R) and IPH are trained on the training data of IntHarmony. While testing with PortraitTest, we compare with two more configurations: Bargain-Net (R) (++) and IPH (++) which are further finetuned on augmented

Table 1: Quantitative comparison in terms of reference based performance metrics with previous methods. We compare our proposed method IPH in terms PSNR, SSIM and CIE Delta with previous methods across 2 test datasets. **Red** and **Blue** corresponds to first and second best results. ++ corresponds to the configuration where the model is further finetuned on augmented MHP dataset.

| Dataset | Type | Method | Venue | PSNR (↑) | SSIM (↑) | CIE Delta (↓) |
|---------|------|--------|-------|----------|----------|---------------|
| PortraitTest | Direct Composite | - | - | 27.21 | 0.9709 | 37.27 |
| | Generic Harmonization | DoveNet Cong et al. (2020) | CVPR 20 | 27.44 | 0.9314 | 23.49 |
| | | BargainNet Cong et al. (2021) | ICME 21 | 28.47 | 0.9364 | 20.95 |
| | | RainNet Ling et al. (2021) | CVPR 21 | 28.55 | 0.9315 | 18.32 |
| | Interactive Harmonization | BargainNet (R) Cong et al. (2021) | ICME 21 | 28.56 | 0.9389 | 19.91 |
| | | BargainNet (R) (++) Cong et al. (2021) | ICME 21 | **30.10** | **0.9787** | **12.31** |
| | | IPH (Ours) | - | 30.86 | 0.9553 | 12.16 |
| | | IPH (++) (Ours) | - | **36.52** | **0.9871** | **4.21** |
| IntHarmony | Direct Composite | - | - | 25.22 | 0.8957 | 54.02 |
| | Generic Harmonization | DoveNet Cong et al. (2020) | CVPR 20 | 26.60 | 0.9011 | 32.45 |
| | | BargainNet Cong et al. (2021) | ICME 21 | **27.94** | 0.9102 | 28.65 |
| | | RainNet Ling et al. (2021) | CVPR 21 | 24.92 | **0.9113** | **20.48** |
| | Interactive Harmonization | BargainNet (R) Cong et al. (2021) | ICME 21 | 27.09 | 0.9078 | 21.52 |
| | | IPH (Ours) | - | **30.22** | **0.9190** | **15.62** |

portrait images from the MHP dataset as explained in Section 3.3. As IntHarmony actually contains objects, people and natural scenes, validating on it helps us prove that the interactive feature helps in even normal harmonization. Validating on PortraitTest helps us prove if our method will work well on real-world high resolution portrait images. It also helps us understand where our method stands when compared to professional editors.

**Referenced Quality Metrics:** We use referenced quality metrics like PSNR, SSIM and CIE Delta to quantitatively compare the performance of IPH with the previous methods. Table 1 consists of the performance comparison on both the PortraitTest and IntHarmony test datasets. We note that IPH achieves the best performance in terms of PSNR, CIE Delta and SSIM when compared to the previous harmonization networks. IPH outperforms all previous methods across all metrics in both datasets. It can be noted that BargainNet (R)/ BargainNet (R) (++) do not perform as well as IPH/IPH (++) even though they are trained with the same pipeline and training conditions of IPH. Prior methods fail as they do not consider arbitrary regions to guide the harmonization and also don't account for luminance matching. We get the best performance as the proposed approach is more effective at improving the performance compared to the previous frameworks.

**Visual Quality Comparison:** In Figure 4, we show results on composite portraits from PortraitTest dataset. It can be seen that the harmonized foreground generated by generic harmonization methods like Dove-Net, Bargain-Net, and Rain-Net is inconsistent with color and luminance conditions when compared to the reference portrait. Also, the predictions of Bargain-Net and Rain-Net fail to harmonize the local lighting conditions present in the composite as they fail to explicitly match the highlight, mid-tone and shadow. Bargain-Net (R) (++) performs interactive harmonization however it still fails to match the contrast of the reference. Our method harmonizes the composite well even considering the local luminance conditions and is close to the annotations done by a professional.

**User Studies:** We conduct two different human subjective reviews to compare the performance of IPH with other methods. First, we select 15 images from the PortraitTest dataset and present the predictions of Dove-Net, Bargain-Net (R) (++), Rain-Net and IPH (++) on a screen. We ask 30 subjects to independently rank the predictions from 1 to 3 based on the visual quality considering if the color and luminance conditions of all the portraits in the image look consistent. We give scores 3,2, and 1 for ranks 1,2 and 3 respectively. Additionally, we also conduct a pair-wise A/B user study. Here, we show the input image and the output of two methods to a user. We then collect these votes and report the preference of the users. We apply the Thurstone Case V analysis to these votes to get the preference values. We also report the Bradley-Terry (BT) score . All these scores have been tabulated in Table 2 where our proposed method is observed to be desired the most.

## 6  DISCUSSION

**Ablation Study:** We conduct an ablation study to analyse the importance of each component of our proposed method. We evaluate these models on the PortraitTest dataset. We start with just the Harmonizer Network (H-Net) without AdaIN layers trained for generic harmonization. Then we add

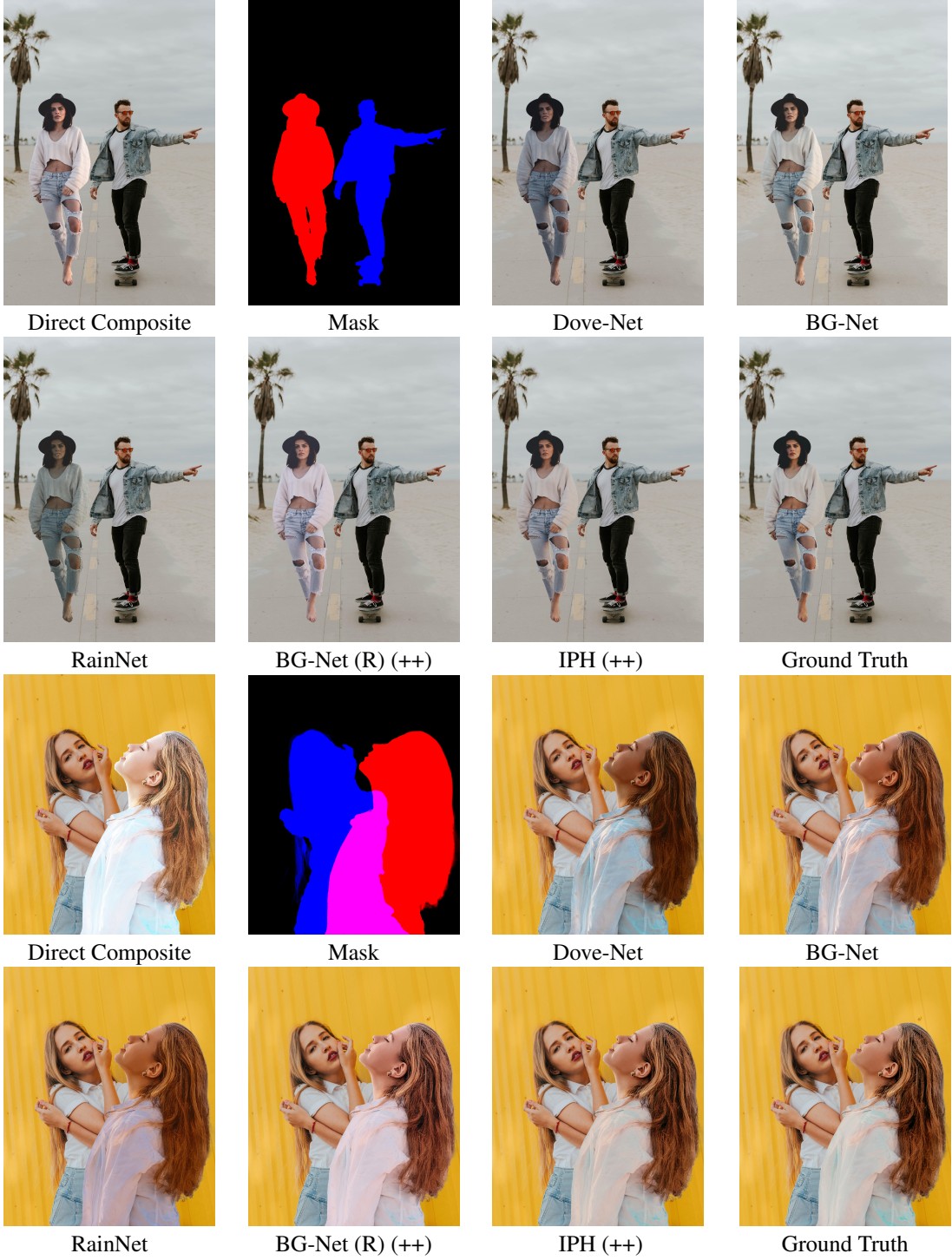

Figure 4: Qualitative comparison on the Portrait Test dataset. Ground Truth annotations here are created by human professional experts. Red mask corresponds to the composite foreground region and blue mask corresponds to the reference region chosen from the background.

Table 2: User Study: Quantitative Comparison with respect to scores from human subjects while evaluating on PortraitTest dataset.

| Method | Ranking Score (↑) | Preference (↑) | BT score (↑) |
|---|---|---|---|
| Composite | - | -0.439 | 0.1577 |
| DoveNet | 19.40 | -0.224 | 0.1602 |
| Bargain-Net (R) (++) | 26.40 | 0.102 | 0.1965 |
| Rain-Net | 12.26 | -0.215 | 0.1600 |
| IPH (++) (Ours) | **28.33** | **0.782** | **0.3252** |

Table 3: Ablation study: We perform an ablation study on real test data to understand the contributions brought about by different techniques proposed in IPH.

| Method | PSNR (↑) | SSIM (↑) | MSE (↓) |
|---|---|---|---|
| Direct Composite | 27.21 | 0.9709 | 155.75 |
| H-Net | 28.40 | 0.9342 | 120.35 |
| w SE | 30.91 | 0.9656 | 53.89 |
| w Mid-tone loss | 31.25 | 0.9715 | 40.54 |
| w Shadow loss | 32.56 | 0.9668 | 50.34 |
| w Highlight loss | 34.21 | 0.9612 | 45.28 |
| w SE + LM loss (IPH) | **36.52** | **0.9872** | **18.33** |

the style encoder and AdaIN layers in the decoder and train for interactive harmonization (H-Net+ SE). All of these methods are trained using the losses $\mathcal{L}_{harmonization}$, $\mathcal{L}_{consis}$ and $\mathcal{L}_{triplet}$. Then we add the novel losses introduced in this work - $\mathcal{L}_{highlight}$, $\mathcal{L}_{mid-tone}$, and $\mathcal{L}_{shadow}$ individually. From Table 3, it can be observed that all the new components introduced in this work are important and play a key role in achieving better performance. Making the framework interactive by using the style encoder and injecting the style code of reference region to guide the harmonization is sufficient the push the performance above previous methods. Furthermore, the proposed luminance matching loss helps improve the performance by a significant margin proving that matching the highlight, mid-tone and shadow of the foreground and reference help obtain better harmonization.

**Robustness to chosen reference region:** One key advantage our framework has over generic harmonization is that it is more flexible and robust. We show that IPH is robust to different regions chosen by the user. We note that the output harmonization adjusts itself with the select region chosen from the background. This is very useful in cases where the luminance conditions vary across different parts of the background image. We illustrate this in Figure 10.

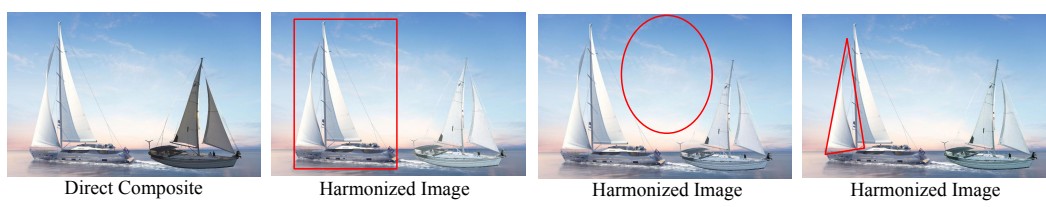

| Direct Composite | Harmonized Image | Harmonized Image | Harmonized Image |

Figure 5: Qualitative results with different regions of background chosen as reference. The first image is the direct composite and the red box in other images show the reference region chosen by the user and the resultant harmonization of the boat can be seen in the respective images.

## 7 CONCLUSION

In this work, we proposed a new framework called *interactive portrait harmonization* where the user has the flexibility of choosing a specific reference region to guide the harmonization. We showed that this helps the user obtain more realistic harmonization results while providing more control and flexibility to the compositing workflow. We also proposed a new luminance matching loss to carefully match the appearance between the foreground composite and reference region. In addition, we introduced two datasets: a synthetic IntHarmony dataset for training and a real-world PortraiTest dataset for testing. Extensive experiments and analysis showed that IPH performs better and is more robust and useful to solve real-world portrait harmonization problems in practical settings compared to the previous methods.

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

## A APPENDIX

**Converting user-guided to automatic?** Please note that although we provide the flexibility for the user to point to a reference region to guide the harmonization, our framework can be made completely automatic. We can make use of saliency detection models Pang et al. (2020) to get the segmentation mask of the most salient object/person in the background and use that as the reference region to guide the portrait harmonization. In fact, we can make this the initial outcome for portrait harmonization and get feedback from the user if they are interested to select a different region or a sub-part of the current region selected to guide the harmonization.

**Different Objects Harmonization:** In IPH, the input to the style encoder does not have any constraint. It need not even be a region present in the background image. Any region from any image can be fed as the input to the style encoder thus making the framework more flexible. This can be useful in the context of style transfer and re-luminance applications. Figure 6 illustrates the harmonization result when the composite foreground is harmonized with a totally different objects from the different reference images.

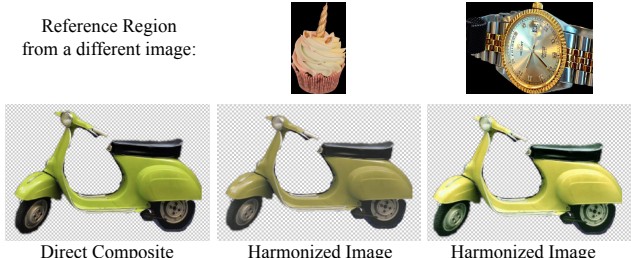

Figure 6: **Qualitative results with reference regions taken from different images.** We show that with IPH we can harmonize the foreground with reference regions present in any image.

**Number of Parameters:** In terms of parameters, IPH is lighter compared to the previous methods. IPH has $27M$ parameters compared to Bargain-Net's $58M$, Dove-Net's $54M$ and Rain-Net's $54M$ number of parameters.

**Tweaking from Harmonization to Color Transfer:** We show that the appearance of the harmonized image can be easily controlled using the style code. If the style code is made to learn the match color instead of harmonizing the image, we show that our framework can be used to perform color transfer. We make two simple changes in our framework to obtain this: (1) increase color augmentations (2) change $\mathcal{L}_{consis}$ loss to make the style codes of the harmonized foreground and the reference region to be far apart from each other; we could obtain style encoder which impart an aggressive color change. We term the style code extracted from this style encoder as $\psi$ and the original style code as $\phi$. Now to control the appearance, we just blend these color codes extracted from these two encoders using different rations. The new style code $\gamma$ would be

$$\gamma = r1\phi + (1 - r2) * \phi, \tag{7}$$

where $r1$ and $r2$ are the blending ratios. In Figure 7, we show the change in appearance for different combinations of $r1$ and $r2$. This shows that IPH can be easily tweaked to make it interactive resulting in realistic harmonization depending on the user's choice of $r1$ and $r2$.

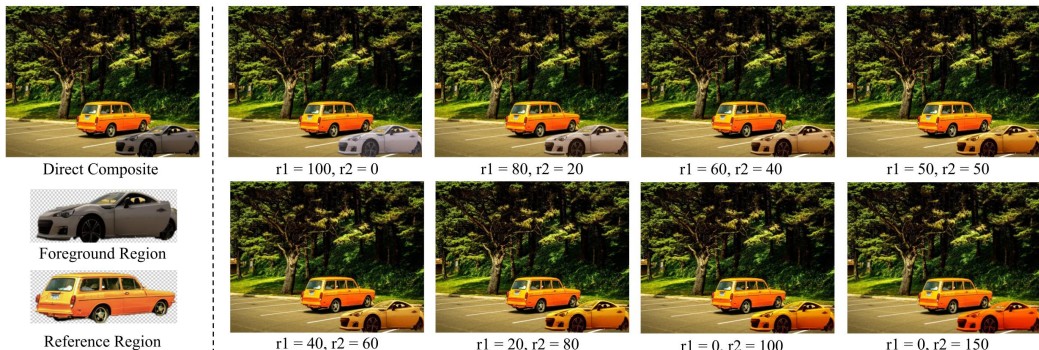

Figure 7: Modifying our framework for color transfer. Qualitative results with different values of r1 and r2 show how we can perform interactive color transfer.

**Limitations:** Although interactive portrait harmonization outperforms previous harmonization methods, there are still some limitations. Our method works fairly well even for in-the-wild composite portraits consisting of people. However, its performance decreases if we test it on portraits consisting of objects. This can be understood as it is still difficult for the style encoder to extract meaningful style code if the object material is peculiar. For example, if the reference object chosen

has a shiny surface or is of a unique material, it is difficult for the style encoder to extract a meaningful style code capturing the texture/material reflectance of the object and use it for harmonization.

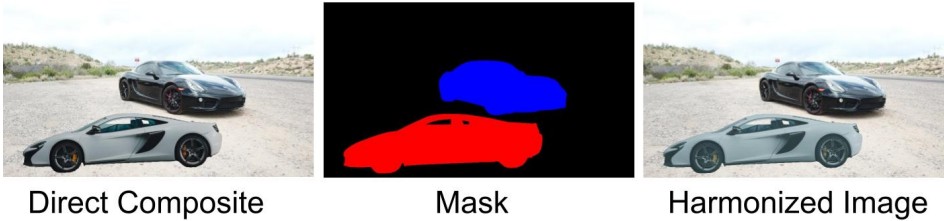

Direct Composite     Mask     Harmonized Image

Figure 8: Limitations: Learning intrinsic material details is difficult in cases where objects are shiny or peculiar.

**Bargain-Net and Rain-Net:** We note that Bargain-Net Cong et al. (2021) and Rain-Net Ling et al. (2021) specifically focus on using the background to guide the harmonization. Bargain-Net has a domain extractor to extract the features of background while Rain-Net proposes a RAIN module which normalizes the background features with the foreground. We tried to make both of these networks interactive by replacing the background mask with a guide mask to select the reference region. With the pre-trained weights, while changing the guide mask we did not see any significant change in terms of performance metrics nor visual quality for both Bargain-Net and Rain-Net. If we retrain the network to take in the guide mask instead of background mask, we noted a small improvement in results for Bargain-Net as reported in the main paper. For Rain-Net, we did not see any improvement. This might be because RAIN module uses the masks in the feature space to normalize and that might not be an optimal way to actually get meaningful features to inject to the foreground. Training it in an interactive way in fact is not robust at all and we get a small decrease in performance. We also not that for these setups changing the guide mask to background mask does not bring out much effect in the harmonized output which makes it clear that these frameworks cannot be converted to serve interactive harmonization efficiently.

**Augmentations:** To generate the composite image in the synthetic dataset IntHarmony, we use a set of augmentations. For each image, these augmentations are chosen at random. To account for appearance changes, we use a brightness and contrast augmentation. We also use color jitter to enforce hue augmentation. In addition, we also use a gamma transformation. Apart from these, we also use 3D LUT augmentations as proposed in Jiang et al. (2021). We also generate local masks and enforce augmentations to cover local lighting effects. Here, we apply augmentations such as soft lighting, dodge, grain merging and grain extraction. Note that all the above set of augmentations are considered as a set and for each image a random augmentation is selected and applied on a composite foreground.

**Potential Negative Societal Impacts:** The proposed method may have negative social impacts if people are not using this model properly. As this work focuses on making an edited image look realistic, it can be used to create realistic fake images. Realistic fake images should be contained as it can act as a tool for harassment and fraud. One potential solution is gated release of trained models.

**More Results:** In Figure 9, we show results on composite portraits from IntHarmony dataset. It can be observed that our method produces realistic harmonization results closer to ground truth when compared to previous methods. The quality of predictions proves the usefulness of our proposed approach to solve the interactive harmonization problem. Please refer to the supplementary material to see many more visualizations.

**In-the-Wild Portrait Harmonization Results:** To further validate our method for in-the-wild photos, we randomly create composite images and check the portrait harmonization results. We visualize these composites and the corresponding results in Figure 5. It can be observed that our method produces realistic harmonization results with the appearance being close to how it would have been if the person was photographed in-situ.

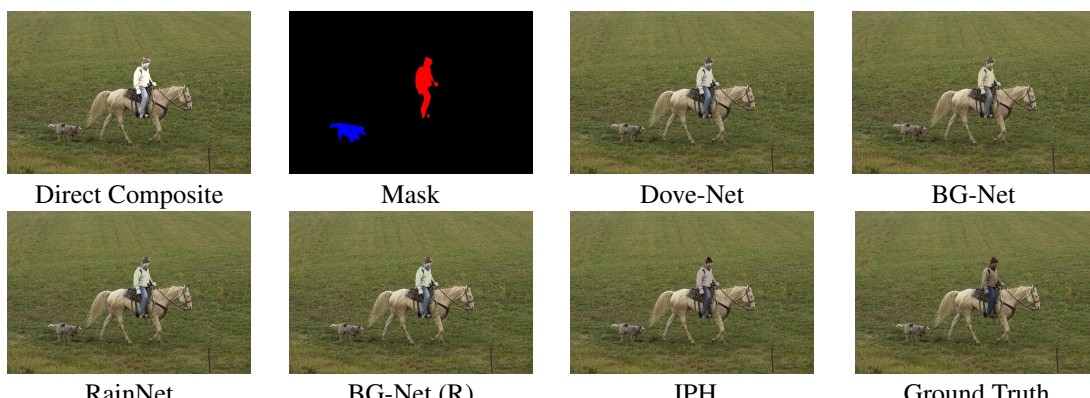

Figure 9: Qualitative comparison on the IntHarmony test dataset. Red mask corresponds to the composite foreground region and blue mask corresponds to the reference region chosen from the background. The other columns correspond to the predictions generated by the corresponding methods. Best viewed zoomed in and in color.

Composite Image     Our Prediction

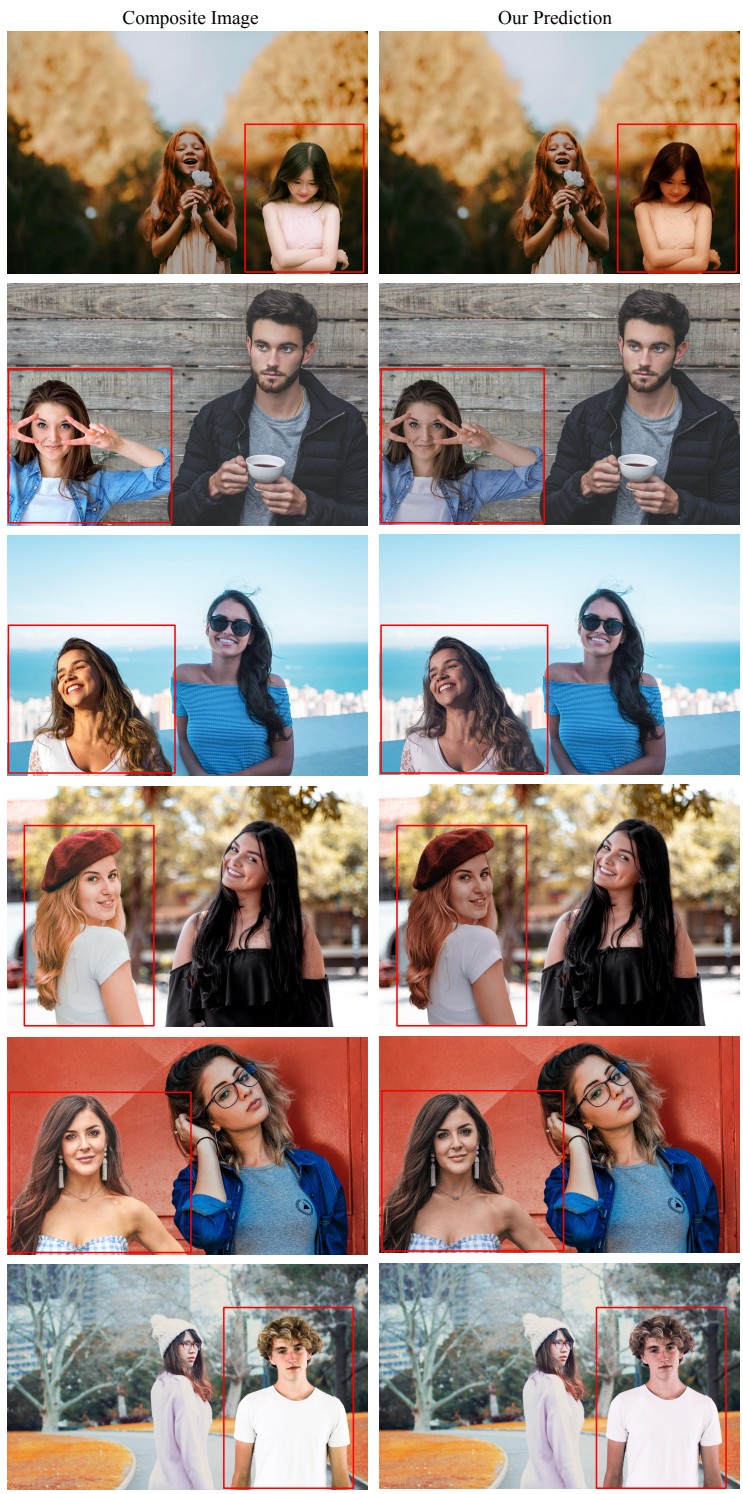

Figure 10: Qualitative results while testing our method for in-the-wild composite portraits. The red box denotes the composite foreground. We use the portrait in background as reference region.

