# OpenReview forum: "Interactive Portrait Harmonization"
_ICLR.cc/2023/Conference — ICLR 2023 poster_

### Official Review · Reviewer_UNoj · 2022-10-17

**Confidence:** 5
**Correctness:** 3
**Technical Novelty And Significance:** 3
**Empirical Novelty And Significance:** 3
**Recommendation:** 8

**Clarity, Quality, Novelty And Reproducibility:**


-> This paper focuses on harmonization but one other task very relevant to it is re-lighting. The paper should discuss more on where this method stands when compared to relighting or why it is different. The proposed luminance loss seems to focus more on the light aspect; so more details  are needed.

-> Comparison with previous methods on “in the wild” edited images for background harmonization is shown only on Fig 1. More examples of how the background harmonization fails would be interesting to see.

->  PortraitTest dataset proposed in the paper is said to have  ground-truth annotations collected from professional experts. There aren't any more details about this. How were these GTs created? Using manual harmonization techniques? If yes, what parameters were changed? More details are needed.

-> The increase in performance seems high for the real dataset when compared to the synthetic dataset which is weird considering the network is trained on synthetic dataset. Any explanations for it?

-> Figure 1 Caption. “Bottom row- Casual portrait.” The term casual can be replaced with Non-studio.


**Strength And Weaknesses:**

-> The  idea of interactive harmonization is interesting and could meet the practical demands of concise human harmonization in the real world. This is the first work to look into this problem.

-> One of the main contributions is a large-scale dataset with reference regions annotated. And they also build a benchmark test with professionally edited harmonization results. This dataset could be helpful if it is being released publicly. The paper has relatively extensive evaluations on the dataset and also vivid examples.

-> The loss function introduced seems to help well in achieving a good performance while looking at the ablation study. The inspiration of the new loss function being what is done in manual harmonization is sound.

->The paper shows results on some “in the wild” edited images which look impressive.

-> The framework is shown to be robust and have different results for different reference regions and also shown to have multiple other use-cases like color transfer and easy switching to automatic harmonization.


**Summary Of The Paper:**

This paper proposes a new framework for interactive harmonization. A new network is proposed to allow users to pick certain regions of the background and perform harmonization on the foreground. A novel luminance matching loss and a new synthetic dataset and a real test dataset is proposed. The proposed network is shown to perform better than previous generic harmonization methods on the new datasets as well as on some “in the wild” edited portrait images.

**Summary Of The Review:**

Based on the strength and weakness mentioned above, I recommend to accept this paper.

---

> ### Author Response · Authors · 2022-11-19
> **Official Response to Reviewer UNoj**
>
> We thank the reviewer for the suggestions. In the following, we give explanations for all the concerns raised by the reviewer:
>
> ### 1) Relighting:
>
> Portrait relighting focuses on changing the lighting characteristics of the portrait to match the approximate realistic lighting effects in a scene. Image harmonization aims to adjust the complete statistics of the foreground with that of the new scene so that the foreground does not look to be edited. The main goal of harmonization is to make sure that the composite foreground looks as if it was in-situ when the image was captured.
> We note that luminance is one main aspect of harmonization however unlike most relighting methods which estimate the geometry, albedo, and lighting and render the image with a new lighting, the harmonization problem is considered as an image-to-image translation problem. This discussion and more related works on relighting can be found in the appendix.
>
> ### 2) More “in-the-wild” Results:
>
> We have added more comparison between background harmonization and interactive harmonization in the supplementary material. It can be observed that our method produces realistic harmonization results with the appearance being close to how it would have been if the person was photographed in-situ.
>
> ### 3) PortaitTest:
>
> The ground truth of PortraitTest dataset is annotated with the help of multiple professionals. Ground-Truth is annotated using PhotoShop curve adjustment layers to change brightness, hue, saturation layer and contrast of the composite foreground to make it look more natural and pleasing.
>
>
> ### 4) Better performance on real dataset:
>
> We thank the reviewer for pointing out this observation. We would like to mention that the real world datasets that we use are all portrait images in which there is a clear distinction between the person in the background and the background itself. For example, harmonizing in-studio portraits with the background does not make sense as the studio screens are of some different color than the lighting condition (as seen in Fig 1). It is in these cases where our proposed method performs really well as one can choose the person in the background instead of the screen in the interactive setup. This explains why our method is better performing at real datasets than synthetic datasets which also highlights the importance of our work.
>
> ### 5) Typo:
>
> We thank the reviewer for the suggestion. We have changed it in the revised edition.

---

### Official Review · Reviewer_MxTM · 2022-10-19

**Confidence:** 5
**Clarity, Quality, Novelty And Reproducibility:** The paper is well organized and appea…
**Correctness:** 3
**Technical Novelty And Significance:** 3
**Empirical Novelty And Significance:** 3
**Recommendation:** 5

**Strength And Weaknesses:**

Strength
+ A novel framework with a luminance matching loss is proposed to solve the interactive portrait harmonization.
+ The paper is well-written and easy to follow.

Weaknesses
- The harmonized foreground is only suitable for the reference region, not the foreground placement region. In fact, in the real world, the illumination at different locations in the background might be different, which results in appearance differences of the foreground object. Besides, is it reasonable to RANDOMLY choose an instance mask for dataset construction as a reference guide mask? How to ensure the luminance/color consistency between the foreground placement location and the random reference location?
- The performance of the interactive image harmonization heavily depends on the user’s selected reference guide region. Compared with those automatic image harmonization methods, the reference guide region selected by the user is not necessarily optimal. In addition, in some cases, the partial region of the background is inadequate to infer the luminance/color conditions of the foreground placement region.
- For the luminance matching loss, the author claimed three new losses are used to match the highlight, mid-tone, and shadow between the reference region and foreground region. But these equations (1)-(3) indicate that the losses only relate to the predicted harmonized image and the ground truth, which is unrelated to the reference region.


**Summary Of The Paper:**

This paper proposed a new framework, namely interactive portrait harmonization, to allow users to select a reference region in the background image to guide the harmonization. The key to this framework is a luminance matching loss to ensure the luminance consistency between the selected reference region and the foreground region. Two datasets are also constructed for training and testing, respectively. Extensive experiments on natural and synthetic data demonstrate the effectiveness of the proposed framework.


**Summary Of The Review:**

The paper introduced a new setting, interactive portrait harmonization, to allow users to select a reference region in the background image to guide the harmonization. However, the luminance/color conditions of the reference region do not necessarily match that of the foreground region, which limits the realism of interactive harmonization. In addition, the reference guide region is subjective and not necessarily optimal. Thus, I give the paper a borderline reject rating.

---

> ### Author Response · Authors · 2022-11-19
> **Official Response to Reviewer MxTM**
>
> We thank the reviewer for the suggestions. In the following, we give explanations for all the concerns raised by the reviewer.
>
> ### 1) Foreground placement region vs Reference region:
>
> We agree with the reviewer that the illumination might be different at different places in the background. To solve exactly this issue, we proposed an interactive framework where the user has the flexibility of choosing which region in the background the foreground should be harmonized with. Please note having different illumination at the background is exactly the problem we solve in this work. The edited foreground can have different lighting conditions depending on where the foreground is placed with respect to the background. Our framework gives flexibility to the user to select which part of the background the new foreground should be harmonized to. So, depending on where the foreground is placed, the user can just pick regions near the foreground placed region or any other region based on their choice to perform the harmonization. This has also been visualized in Figure 5.
>
> ### 2) Is it reasonable to RANDOMLY choose an instance mask?
>
> In portrait editing, one usually places the composite foreground image at some location in a background image. In most portrait cases, the composite will need to be harmonized with the person/people in the background to make the edited image look real. From this observation, we choose to pick a random instance mask in the background to choose as our reference region. We do agree that there might be few cases where this might not be optimal. However, from observing the dataset and the instance masks, we concluded that in most cases this choice is optimal and manually choosing the best instance mask is a humongous task. So, we just picked random instance masks in the background to choose as the reference region and notably, we were able to obtain good results with it further validating our choice.
>
> ### 3) What if the region chosen by the user is not optimal?
>
> For this question, we would like to mention that having a rigid framework where the user has no control of how the harmonization is done is worse than giving the user complete control of how the image should be harmonized. Picking the right reference region for harmonization is entirely upon the user and is usually simple in Portrait editing. Most studio portraits contain only a few people on the scene and so it is very easy for the user to pick the optimal reference region (a portrait in the background). To understand this, please check Figure 1 and 10. In these studio portraits, the number of people found are just 2 and the user normally picks the portrait in background as reference.
>
> Even if the user is not sure how to pick the optimal region, they can always pick the entire background as reference similar to how harmonization is done in current setups. Either way, our framework is more useful than rigid automatic harmonization setups that exist.
>
> Also the reviewer mentions that “in some cases” the partial region of the background is inadequate to infer the luminance/color conditions of the foreground placement region. We agree that such cases do exist in the real world but in our framework the user also has the flexibility to choose the entire background as reference in those cases. Please note that it is not a necessity that the reference needs to be a region of the background, it can simply be the complete background too. So in those cases, the user can choose the entire background region instead of a smaller region of background.
>
> ### 4) Equations (1) to (3):
>
> We apologize for the mistake and thank the reviewer for pointing this out. This was a writing mistake. Equations (1)-(3) are between the predicted harmonized image and the reference image. The change has been rectified in the revised paper.

---

> > ### Comment · Reviewer_MxTM · 2022-11-21
> > **Re: Official Response to Reviewer MxTM**
> >
> > After reading the authors' response, my concerns remained.
> >
> > Since the authors agree that the illumination may be different at different places, is the illumination near the foreground region the same as that in the foreground region? The critical issue of image harmonization is how to infer *the illumination/color conditions of the foreground region* given a foreground region and a background image. However, the proposed interactive harmonization avoids this issue and leaves it to the user.
> >
> > In addition, the background image in Figure 5 is from an outdoor scene. From the shadows on the boat in the reference region, it can be inferred that the sun is roughly located in the lower-left corner of the image. However, there is no shadow generation on the harmonized boat, just the brightness/color changes. Many examples in the paper are under low-frequency/smooth illuminations. What if it's under high-frequency illumination? For example, What if the background of the first example in Figure 4 is from a sunny outdoor scene?

---

> > > ### Author Response · Authors · 2022-11-23
> > > **Response to Reviewer MxTM**
> > >
> > > We are sorry that the reviewer's concerns remain even after our response. We hope this reply is more clear and solves the reviewer's concerns.
> > >
> > > ### 1)  Is the illumination near the foreground region the same as that in the foreground region?
> > >
> > > This is a subjective question and the characteristic of the illumination near the foreground region really depends on the scene. For example, if there are a lot of local illumination changes near the foreground region, the foreground need not necessarily have the same illumination as its nearby region. The proposed interactive harmonization does not avoid this issue but actually gives the user the flexibility of choosing which region (near the foreground/ some background region) the foreground should be harmonized with. This flexibility can help the users choose specific regions with which the input needs to be harmonized with unlike automatic frameworks which just pick the full background.
> > >
> > > ### 2) Figure 5:
> > >
> > > We agree with the reviewer that the shadows of the left boat are not that well transferred to the right boat in Figure 5. Even though we use a reference region based contrast loss, the ground truth L1 loss does not have these local changes. This causes these small local characteristics not transferred well to the foreground. We will be exploring a better reference based loss to capture such local change in the future.
> > >
> > >  ### 3) High-frequency illumination:
> > >
> > > We request the reviewer to look at the supplementary material where we illustrate cases show sharp changes in luminance between foreground and background. For example, in Figure 3 (top and bottom), the composite foreground is high frequency illumination. Our method handles this case well and harmonizes it well. Also, in Figure 2 (top), the reference region is of high frequency illumination and our proposed method handles this setup too quite well.

---

### Official Review · Reviewer_DBoQ · 2022-10-24

**Confidence:** 4
**Correctness:** 4
**Technical Novelty And Significance:** 3
**Empirical Novelty And Significance:** 3
**Recommendation:** 6

**Clarity, Quality, Novelty And Reproducibility:**

The paper is well-written besides a typo of "a interactive fashion" on page 6. The novelty of selecting a region as a reference area may not be very strong because it would be user interface issue rather than research issue of image harmonization.

**Details Of Ethics Concerns:**

No concerns.

**Strength And Weaknesses:**

Strength

1. Experimental results show the effectiveness significantly.
2. The ablation study shows the effectiveness of using the proposed luminance matching loss.

Weakness
1. In term of deep learning model, it looks like very similar to the previous model of DoveNet. The paper does not discuss the difference in details.

2. If two objects are in different colors, such as red car as a reference region and blue car as a foreground image, what would happen?

**Summary Of The Paper:**

The paper introduces interactive image harmonization that enables the user to select a specific object/person to guide. The paper also proposes a new technique of using luminance matching loss and a new dataset for validating portrait harmonization. Experimental results show the effectiveness over the previous methods.

**Summary Of The Review:**

The paper is introducing interactive image harmonization that allows the user to select a region as a guide. When two or more people/objects are in the same image, if the user uses a person/object as a reference image, the generated image would be better especially for portrait photos taken in a studio with a solid background. Experimental results show the effectiveness of the proposed method.

---

> ### Author Response · Authors · 2022-11-19
> **Official Response to Reviewer DBoQ**
>
> We thank the reviewer for the suggestions. In the following, we give explanations for all the concerns raised by the reviewer.
>
> ### 1) Difference with DoveNet:
>
>  DoveNet uses attention-enhanced U-Net with a domain verification network trained in an adversarial way. It takes in the edited image and predicts a harmonized image. Our proposed IPH framework differs from DoveNet framework according to the following points:
>
> i) DoveNet has no provision in its framework to harmonize the foreground with a select background while IPH has a separate branch to incorporate a reference region.
> ii) IPH has a style encoder which extracts a style code from DoveNet. This is not present in DoveNet.
> iii) IPH uses adaptive instance layer normalization layers in the decoder to copy the style code of the reference region to the original image. This is not present in DoveNet.
> iv) Our deep learning model also does not incorporate a discriminator nor domain verification modules like DoveNet.
>
> ### 2) Color Change:
>
> If we use a red car to harmonize a blue car, in the original implementation of IPH the color will not change. The blue car will be harmonized to the brightness and contrast values of the red car but the hue will not change as we have trained the network in such a way. However, if one wants to incorporate color change, it is possible to do it with our proposed framework with a simple change.  If the style code is made to learn the match color instead of harmonizing the image, our framework can be used to perform color transfer. We show how to do this and also show some results in the appendix of the paper (Figure 7).
>
> ### 3) On interactive framework:
>
> We agree with the reviewer that adding the interactive part is kind of a user interface issue. However, we would like to emphasize that just changing the user interface will not solve the issue. The way we develop and infer the model needs to change to incorporate interactive behavior. This needs complete modification of the framework and also the training and testing datasets. Our paper analyzes this problem and proposes a new framework, loss, and datasets to solve the interactive harmonization problem.
>
> ### 4) Typo:
>
> We thank the reviewer for pointing out the typo. We will change it accordingly in the revised version.

---

### Official Review · Reviewer_53Gf · 2022-10-24

**Confidence:** 5
**Correctness:** 3
**Technical Novelty And Significance:** 3
**Empirical Novelty And Significance:** 3
**Recommendation:** 6

**Clarity, Quality, Novelty And Reproducibility:**

Clarity:

- The paper has not been fully edited. For example, section 2 is almost impossible to read, as the name of the authors in many papers is repeated twice (included in the text, and also as the reference). Also, Table 2 does not make sense if the ranking are going between 1 and 3; I am guessing the authors put the decimal point in the incorrect location.

Quality:
- The paper seems to have enough quality for ICLR. The idea is good and the results improve over the state of the art.

Novelty:
- The paper seems to be novel enough; the idea of interactive harmonization deserves publicatiion.

Reproducibility:
- Authors will publish all the codes once the paper is accepted.

**Details Of Ethics Concerns:**

Authors need to better explain all the GDPR concerns about PortraitTest, specially from where the original portraits are taken.

**Strength And Weaknesses:**

Strenghts:

- The paper is easy to follow.
- The results overtake the state-of-the-art.
- Allowing the user to select the region of interest is interesting.

Weaknesses:
- Luminance Matching Loss: The Luminance matching loss is not perceptual. It is well known that to be able to perceive a change of luminace value, we need a much larger difference in the highlight regions that in the shadow regions (Weber's law). For this reason, I stringly believe than the three different terms in the luminace loss shall be weighted according to this fact.

- The authors use a ranking experiment for the user study. These type of experiments are known to be very noisy. Also, there is not a reference in how statistical significant is the distance of 0.19 between IPH and Bargain-Net. For this reason, the authors should instead go for a pair-wise comparison, and then use the Thurstone Case V law (See [1]) to also study the statistical significance of the results.

- Authors need to discuss why when including the Shadow loss and the Highlight loss the method improves PSNR but get worse in SSIM. I believe it might be related to my first comment.

- Also, it is not make sense to put both PSNR and MSE in the Tables. They are completely correlated as PSNR is directly computed from MSE. Also, authors need to include some Color metric, in order to prove that no color artifacts, such as Hue shifts appear in the result. I recommend CIE DElta E.

[1]  Color Gamut Mapping, Ján Morovič, 2008, John Wiley & Sons, Ltd


**Summary Of The Paper:**

The authors propose a new image harmonization method. Image harmonization aims at compositing regions of different images in a way the observer can not tell they were coming from different original images. The main novelty of this work is the allowance of interaction by the user. In the proposed model the user can select which part of the background image should be used for harmonization. Furthermore, thae authors also propose a new loss based on luminance inspired by professional users.

**Summary Of The Review:**

This is an interesting paper with good results. Some issues need to be addresses (see my detailed explanation in the Weaknesses section). For this reason, my rating is somehow between 5 and 6.

---

> ### Author Response · Authors · 2022-11-19
> **Official Response to Reviewer 5Gf**
>
> We thank the reviewer for the comments and suggestions. In the following, we give explanations for all the concerns raised by the reviewer.
>
> ### 1) A/B user study:
>
>  As per reviewer’s suggestion, we have conducted a pair-wise A/B user study. Here, we show the input image and the output of two methods to a user. We then collect these votes and report the preference of the users. We apply the Thurstone Case V analysis to these votes to end up with the preference values. We also report the Bradley-Terry (BT) score which is a statistical technique that prioritizes rankings of a list of attributes into importance for the same attributes on a powerful ratio scale. In both these metrics, we observe our method to perform better than the previous methods under comparison. The new user study table can be found below:
>
>
> | Method     | Preference | BT Score |
> |------------|------------|----------|
> | Composite  | -0.439         | 0.1577        |
> | DoveNet    | -0.224            | 0.1602         |
> | BargainNet | 0.102            |  0.1965        |
> | RainNet    |  -0.215          |  0.1600        |
> | IPH (Ours) |  0.782          |  0.3252        |
>
> ### 2) Luminance Matching Loss:
>
> We thank the reviewer for the suggestion and we agree that weighing the highlight, mid-tone and shadow losses before adding them is a good idea. However, we would like to point out that we are actually using the lightness from CIE Lab instead of luminance in the proposed loss, which should be already accounting for some of those perceptual issues. Additionally, our highlight, mid-tone and shadow points are relative to the L histogram of each region (90th, 50th and 10th percentiles respectively). So in a sense, our proposed loss is enforcing a perceptual simplified histogram matching. While more complex perceptually-motivated losses could be developed, the proposed one is simple and seems to work well enough for our purposes.
>
> ### 3) Trends of PSNR and SSIM:
>
> We note that SSIM incorporates important perceptual phenomena, including both luminance masking and contrast terms. So from Table 5, when we look at just H-Net and SE encoder the SSIM is actually lower than the composite image. This tells us how the luminance and the contrast characteristics of the harmonized image with just the network change is not optimal. Once we add mid-tone, shadow and highlight; we get better SSIM than just H-Net or H-Net with SE. This shows us that having these losses actually improves the perceptual quality of the image.
>
> Also, please note that in Table 5 ablation results, the rows correspond to losses added individually. This means that the row corresponding to Mid-tone is the configuration of H-Net with Mid-tone loss and the row corresponding to Highlight is the configuration of H-Net with Highlight loss. The improvement in SSIM over H-Net for high-light is more than that of mid-tone or shadow, but we see improvement in SSIM for all these losses over H-Net. This shows that each of the individual loss components brings in a boost in performance. Adding everything together brings in the best boost. While we agree that more optimization could be done by weighing them, the proposed one is simple and seems to work well enough for our purposes.
>
> ### 4) Addressing Hue shifts:
>
> We have added the comparison of CIE Delta E and have revised our table. We have also removed MSE as per reviewer’s suggestion. We notice that our proposed method gets the best CIE delta score compared to all methods we compared against. We have updated them in the revised edition. The CIE Delta scores are tabulated in the table below for the PortraitTest dataset:
>
> | Method     | PSNR | SSIM | CIE Delta |
> |------------|------------|----------|----------|
> | Composite  | 27.21         | 0.9709       | 37.27
> | DoveNet    | 27.44          | 0.9314        | 23.49
> | BargainNet |  30.10           |  0.9787        | 12.31
> | RainNet    |  28.55         |  0.9315        | 18.32
> | IPH (Ours) |  036.52          |  0.9871        | 4.21
>
>
> ### 5) Editing:
>
> We have thoroughly edited Section 2 and also have added more explanations and removed typos.
>
> ### 6) Data Source and Legal Compliance:
>
> The dataset was acquired from an internal source. Further details about the source would breach the anonymity of the authors. We would make sure to add more details about the same in the final version. Legal compliance was obtained from the subjects for the images to be used for research purposes.

---

> > ### Comment · Reviewer_53Gf · 2022-11-28
> > **Rebuttal read**
> >
> > This rebuttal has solved most of my issues. I believe the paper can be accepted.

---

### Decision · Program_Chairs · 2023-01-20

**Decision:**

Accept: poster

**Justification For Why Not Higher Score:**

It's a good paper, but not groundbreaking.

**Justification For Why Not Lower Score:**

the paper seems technically sound, introduces a new potentially practically relevant problem setting, a corresponding dataset, and a working solution

**Metareview: Summary, Strengths And Weaknesses:**

The paper proposes an approach for portrait harmonization - adjusting the appearance of a foreground object pasted into a photo, to make for a realistically looking composite. In particular, the paper studies a new setting, interactive harmonization, where not the full background image is used as a reference for harmonization, but a (user-selected) region. The method seems to work well and outperform the alternatives.

The reviewers are overall quite positive about the paper, with three of them arguing for acceptance and one against. The main points are as follows:

Pros:
1. An interesting and new setting
2. Good experimental results, including comparisons to baselines and an ablation study
3. A new dataset that will be released

Cons:
1. Somewhat limited technical contribution
2. Doubts about random selection of reference mask being a good choice and generally putting work of reference selection on the user being a good idea

Overall, the paper seems technically sound, introduces a new potentially practically relevant problem setting, a corresponding dataset, and a working solution. So I recommend accepting the paper. I encourage the authors to address the reviewers' comments where possible, in particular perhaps run some quantitative experiments on the effect of region selection on the result.

**Note From Pc:**

if the above contains the word "oral" or "spotlight" please see: "oral" presentation means -> notable-top-5% and "spotlight" means -> notable-top-25%. As stated in our emails, we are disassociating presentation type from AC recommendations